# The Pleiotropic Ubiquitin-Specific Peptidase 16 and Its Many Substrates

**DOI:** 10.3390/cells12060886

**Published:** 2023-03-13

**Authors:** Jiahuan Zheng, Chunxu Chen, Chunqing Guo, Cody Caba, Yufeng Tong, Hengbin Wang

**Affiliations:** 1Department of Internal Medicine, Division of Hematology, Oncology, and Palliative Care, Massey Cancer Center, School of Medicine, Virginia Commonwealth University, Richmond, VA 23298, USA; 2Department of Obstetrics and Gynecology, Li Ka Shing Faculty of Medicine, The University of Hong Kong, Hong Kong 999077, China; 3Department of Bioengineering, School of Engineering, Virginia Commonwealth University, Richmond, VA 23298, USA; 4Department of Human and Molecular Genetics, School of Medicine, Virginia Commonwealth University, Richmond, VA 23298, USA; 5Department of Chemistry and Biochemistry, University of Windsor, Windsor, ON N9B 3P4, Canada

**Keywords:** ubiquitin-specific peptidase 16, histone H2A, substrates, structure, human diseases

## Abstract

Ubiquitin-specific peptidase 16 (USP16) is a deubiquitinase that plays a role in the regulation of gene expression, cell cycle progression, and various other functions. It was originally identified as the major deubiquitinase for histone H2A and has since been found to deubiquitinate a range of other substrates, including proteins from both the cytoplasm and nucleus. USP16 is phosphorylated when cells enter mitosis and dephosphorylated during the metaphase/anaphase transition. While much of USP16 is localized in the cytoplasm, separating the enzyme from its substrates is considered an important regulatory mechanism. Some of the functions that USP16 has been linked to include DNA damage repair, immune disease, tumorigenesis, protein synthesis, coronary artery health, and male infertility. The strong connection to immune response and the fact that multiple oncogene products are substrates of USP16 suggests that USP16 may be a potential therapeutic target for the treatment of certain human diseases.

## 1. Introduction

Ubiquitination is a reversible post-translational protein modification that mediates various downstream signals and plays critical roles in many cellular processes, canonically, protein quality control [1]. The attachment of ubiquitin (Ub) protein to a substrate requires three steps involving three enzymes: Ub-activating enzyme (E1), Ub-conjugating enzyme (E2), and Ub ligase (E3), resulting in the formation of a native or isopeptide bond between the Ub *C*-terminus and a substrate’s *N*-terminal α-amine (Met1) or internal lysine ε-amine, respectively [2]. Ubiquitination either alters protein characteristics or tags them for proteasome-mediated degradation. To counteract ubiquitination, the human genome encodes ~100 deubiquitinases (DUBs) [3]. These DUBs remove Ub from substrates and are essential for maintaining the homeostatic levels of cellular ubiquitination. DUBs can be subdivided into seven families: ubiquitin-specific peptidase (USP), ubiquitin *C*-terminal hydrolase (UCH), ovarian tumor protease (OTU), Machado-Joseph domain-containing protease (MJD), motif interacting with Ub-containing novel DUB family (MINDY), zinc finger containing ubiquitin peptidase 1 (ZUP1), and Jab1/Mov34/MPN^+^ (Mpr1 Pad1 *N*-terminal^+^) domain protease (JAMM) [3]. The first six families are cysteine-dependent proteases, while the JAMM family is composed of zinc-dependent metalloproteases.

The USP family constitutes the largest family of DUBs, with about 58 members; however, this number grows significantly when accounting for alternative isoforms, especially the highly similar USP17-likes gene [4,5,6,7]. USPs share a structurally homologous catalytic domain (CD) but are diverse in their composition of accessory domains that are critical for protein-protein interactions and activity regulation via inter- and intramolecular allosteric mechanisms [3,8]. These DUBs are generally regarded as linkage unspecific, meaning they hydrolyze many or all poly-Ub topoisomers (M1, K6, K11, K27, K29, K33, K48, and K63-linked) in a variety of contexts [9,10]. However, it is becoming apparent that USPs display linkage preferences [11,12], a characteristic explored to uncover these critical enzymes’ cellular function.

Ubiquitin-specific peptidase 16 (USP16), initially known as Ubp-M, was identified as an in vitro histone H2A DUB that undergoes phosphorylation as cells enter mitosis and dephosphorylation during the metaphase/anaphase transition [13]. It was later confirmed to be the major histone H2A DUB responsible for removing Ub from Lys119 (H2AK119ub) [14]. Additionally, H2AK13/15 ubiquitination, which is a modification associated with DNA damage repair [15], is also reversed by USP16 [16]. Cyclin-dependent kinase-1 (CDK1) was shown to phosphorylate USP16 at Ser552, retaining it in the nucleus and enabling proper cell cycle progression and proliferation [17]. Indeed, the knockdown of USP16 in HeLa cells attenuates cell growth due to a disrupted mitotic cell cycle [13,14,17]. By deubiquitinating H2Aub, USP16 facilitates the phosphorylation of histone H3 Ser10 and chromosome segregation at the onset of mitosis [14]. As a potential antagonist of H2A ubiquitination by Polycomb Repressive Complex 1 (PRC1), USP16 deubiquitinates H2Aub at promoter regions. It relieves its repressive effects on *Hox* and many other development-related genes in mouse embryonic stem cells, somatic stem cells, and progenitor cells [18,19,20]. These genes are critically required to maintain these cells’ proper function and identity [18,19,20]. Consistently, knockout of *Usp16* results in mouse embryonic lethality, and overexpression of USP16 in the humanized mouse model of Down syndrome reduces the self-renewal of hematopoietic stem cells and the expansion of mammary epithelial cells, neural progenitors, and fibroblasts [18,19]. While the best-characterized function of USP16 is acting as a histone H2A deubiquitinase in the nucleus, the majority of USP16 is localized to the cytoplasm [13,17,21]. Recently, many new substrates of USP16 have been identified, extending its regulation to diverse physiological and pathological processes. This review summarizes recent advances in USP16 studies, highlighting its potential role in cancer and immune response.

## 2. Structure, Expression Regulation, and Subcellular Localization of USP16

### 2.1. Protein Structure of USP16

#### 2.1.1. Domain Architecture and Conservation

Human USP16 is an active DUB of 823 a.a. It features a UBP-type zinc-finger domain (ZnF-UBP) at the *N*-terminus (a.a. 23–146) followed by a catalytic domain (CD, a.a. 193–823) that is bifurcated by a long, disordered insertion (a.a. 393–627) of about 230 residues (Figure 1A). The closest paralog of human USP16 is USP45, which has 38% sequence identity and the same domain architecture. USP45 has been found to play a role in DNA damage repair [22], but it has not been linked to histone deubiquitination. To understand the relationship between USP16 and histone H2A deubiquitination, we searched for putative USP16 orthologs from representative species of invertebrates, vertebrates, and plants. Notably, budding and fission yeasts (*Saccharomyces cerevisiae* and *Schizosaccharomyces pombe*) do not have an apparent ortholog for USP16. Orthologs of both USP16 and USP45 can be found in vertebrate genomes. Still, there is only one USP16 ortholog from *Drosophila melanogaster*, annotated as Usp16-45, which is highly divergent from orthologs from other species (Figure 1B and Figure 2), with a 26.9% sequence identity to human USP16. However, USP16 orthologs in the *Arabidopsis thaliana* genome are less obvious. It has been previously shown that *Arabidopsis thaliana* does contain H2Aub and Polycomb Group (PcG) protein homologs [23,24]. One reference protein (NP_567705, UBP16), despite being annotated as plant Usp16, contains a much shorter MYND-type ZnF domain instead of a UBP-type ZnF domain. Using the human USP16 ZnF-UBP domain as the query sequence, we identified two plant gene products as the closest putative homologs, UBP1 (NP_565753) and UBP2 (NP_563719), with 24.0% and 25.9% sequence identity, respectively. In comparison, plant UBP16 has only 17.1% sequence identity. We thus designate UBP1 and UBP2 as the *Arabidopsis thaliana* orthologs of human USP16. Such designation is purely based on bioinformatic analysis and requires future experimental evidence for support. The fact that fly has only one ancestor Usp16-45 gene but higher organisms, including plants, have two genes, USP1, and USP45, suggests a gene duplication event during evolution involving USP16, rendering USP16 fine-tuned for its functional roles. Indeed, it has been suggested USP16 and USP45 may arise from a novel whole genome duplication event [25]. Comparing the sequence identity of different parts of USP16, the core CD and ZnF-UBP domain are more conserved than the rest of the protein (Figure 1B), suggesting it likely employs similar mechanisms for substrate recognition but may have different modes of regulation involving the disordered regions. Except for mouse and human USP16, the function of orthologs in other organisms is yet to be investigated.

#### 2.1.2. Overall Structure

Our previous study demonstrated that USP16 forms a tetramer in solution [14]. The size of the protein and the long predicted disordered insertion within the CD render it recalcitrant for structure determination by NMR or X-ray crystallography. Only an NMR structure of the ZnF-UBP domain has been experimentally determined [26]. We resort to AlphaFold [27] for the structure of full-length USP16. AlphaFold-multimer [28] predicts a tightly packed tetrameric structure where the Ub-binding pocket of the CD is occupied by other subunits, which is not physiologically meaningful. Therefore, without further experimental data, we limit our discussion to the monomeric structure of USP16 predicted by AlphaFold (Figure 3). As expected, the predicted structure forms two well-defined domains (ZnF-UBP and USP) of relatively high confidence, along with long disordered insertions. A lysine-rich region (a.a. 437–464) in the middle of the insertion is predicted to form an α-helix with relatively high confidence (Figure 3A).

#### 2.1.3. ZnF-UBP Domain

The experimentally determined ZnF-UBP structure matches the AlphaFold-predicted structure very well. It adopts a two-layer sandwich containing four α-helices surrounding a five-stranded β-sheet. The ZnF-UBP chelates three zinc ions in a cross-braced fashion. The ZnF-UBP domain has been shown to bind the free C-terminal tail of Ub with micromolar affinity [26]. Furthermore, an enzymatic assay [8] using ubiquitin C-terminal 7-amido-4-methylcoumarin (Ub-AMC) as a substrate reveals that full-length USP16 has a 3.5-fold smaller Michaelis-Menten specificity constant (*k_cat_*/*K_M_*) than the CD, suggesting the ZnF-UBP domain plays a regulatory role, although the molecular details of such remain elusive.

#### 2.1.4. Catalytic Domain

In the AlphaFold-predicted structure of full-length USP16, disordered residues (a.a. 571–604) in a segment of low-confidence prediction occupy the Ub binding pocket loosely (Figure 3A). Without experimental evidence, we have no reason to suggest this is the actual case under physiological conditions. We limit the discussion of the CD structure to the core domain without the disordered regions. The CD of USP16 adopts a typical hand-like USP fold with palm, thumb, and fingers subdomains. Compared to the prototypic structure of the CD from USP7 (PDB: 1NB8, Figure 3B), the CD of USP16 contains extra structural elements in the thumb subdomain: a β-hairpin (a.a. 231–251) insertion and an extended helix-turn-helix motif insertion in the thumb subdomain. USP16 also contains an extra short α-helix at the tip of the fingers subdomain formed by a 17 residue insertion (Figure 3B).

USPs are cysteine proteases that feature a catalytic Cys-His-Asn/Asp triad. However, USP16 is one of three USPs that utilizes a Ser instead of an Asn/Asp (Cys-His-Ser), the other two being USP30 and USP45, the closest USP16 paralog [30]. The AlphaFold predicted USP16 structure confirms that residues Cys205, His758, and Ser797 form a catalytic triad (Figure 3C) and are poised in a productive conformation, meaning a properly arranged hydrogen-bonded network is present to enable the deprotonation of the cysteine thiol group for nucleophilic attack. Note that a neighboring Asp798 is not utilized as a catalytic residue (Figure 3C).

#### 2.1.5. Disordered Region

The disordered region of a protein is often involved in post-translational modifications and protein-peptide interactions. Recently, a conserved CRM1-dependent nuclear export signal (NES, a.a. 572–581) and a conserved non-canonical nuclear localization signal (NLS, a.a. 437–459) were identified within the long insertion of the CD [21]. The NLS motif binds to the RPS27a subunit of the pre-40S ribosome to promote the maturation of the 40S ribosomal subunit. It is interesting to observe that the predicted structure of this motif is an α-helix [31]. Hence, although the long insertion within the CD is not required for its catalytic activity, it most likely plays important regulatory roles in the subcellular localization and substrate recognition of USP16.

Although USP16 contains multiple predicted CDK phosphorylation sites (a.a. 97, 146, 189, 217, 277, 552, and 600), we experimentally identified phosphorylated Ser330, Ser415, and Ser552 [17]. Both Ser415 and Ser552 are located on the disordered insertion, while Ser330 is located in the thumb subdomain (Figure 3A) and distant from the Ub binding pocket. Phosphorylation of these Ser residues is unlikely to affect substrate Ub recognition. We further demonstrated that Ser552 is the major phosphorylation site, and its phosphorylation allows the nuclear import of USP16, which correlates with H2A deubiquitination at the onset of mitosis [17]. Thus, USP16 pSer552 is specifically linked to mitotic progression. Notably, Ser552 is not conserved in mice, zebrafish, or fruit flies (Figure 2). Thus, these organisms may employ distinct mechanisms for the global H2A deubiquitination during the M phase. The enzymes responsible for Ser330 and Ser415 phosphorylation have not yet been defined. The linker between the ZnF-UBP and USP domains of USP16 (a.a. 137–196) are mainly disordered based on AlphaFold prediction (Figure 3A). We previously found the region interacts with a HECT-type E3 Ub ligase HERC2 [16]. The formation of a DUB/E3 complex is a common theme found throughout the Ub proteasome system. The effect of USP16/HERC2 interaction on the activity of USP16 remains to be elucidated.

### 2.2. Mechanisms Regulating the mRNA and Protein Levels of USP16

#### 2.2.1. Gene Copy Number

In the human genome, *USP16* is located on chromosome 21. Triplication of part or the entirety of chromosome 21 causes Down syndrome [32]. Furthermore, triplication of *USP16* increases the mRNA and protein levels by 0.5-fold [18]. This relatively minor increase has been implicated in the defects of multiple somatic cell lineage and progenitor cells in Down syndrome [18]. RNA interference, deletion, or inactivation of USP16 in one of the triplicated copies can mitigate these stem cell defects and may represent alternate approaches to ameliorate these disorders [18,33,34].

#### 2.2.2. NF-κB (Nuclear Factor Kappa-Light-Chain-Enhancer of Activated B Cells)

Three NF-κB binding sites were identified in the promoter region of USP16 [35]. Overexpression of p65 (also called RelA), one of the NF-κB family of transcription factors, significantly increases USP16 transcript levels. Strong activators of the NF-κB pathway, such as lipopolysaccharide (LPS) and tumor necrosis factor-alpha (TNF-α), also positively regulate USP16 transcription. Therefore, the transcription of USP16 is under the control of the NF-κB pathway [35]. Interestingly, USP16 also regulates the NF-κB pathway (Figure 4A) in a feedback loop. Mass spectrometry analyses identified USP16 as an interacting protein of IKKβ (inhibitor of nuclear factor kappa-B kinase subunit beta), the key kinase of the NF-κB transcription factor p105 [36]. IKKβ is ubiquitinated at Lys238, and USP16 reverses this modification. USP16 DUB activity enhances IKKβ interaction with p105 and promotes its phosphorylation [36]. Consequently, USP16 is critical for activating NF-κB-targeted genes, including USP16 itself (Figure 4A). This positive feedback mechanism may allow USP16 and the NF-κB pathway to adapt the immune system to stimuli or stress more quickly.

#### 2.2.3. Gene Fusion

Chronic myelomonocytic leukemia (CMML) is a heterogeneous hematopoietic disease exhibiting either myeloproliferative or myelodysplastic pathological properties [37]. In a study involving 30 CMML samples from 29 patients, one contained an inversion of chromosomal 21q21-22 [38]. This chromosomal translocation resulted in the fusion of RUNX1 and USP16 genes, containing exon 1 of USP16 fused to exon 15 of RUNX1 and lacking a start codon. However, the fusion gene does contain multiple ATG codons in exons 5–7 in RUNX1 [38]. While this fusion disrupts both the USP16 and RUNX1 genes, the potential for generating truncated RUNX1 protein and its implication to CMML pathogenesis has not been explored.

#### 2.2.4. HBV X (HBx) Protein

Hepatitis B virus (HBV) infection is a major cause of hepatocellular carcinoma (HCC). The major pathogenic protein of HBV is HBV X (HBx) protein, which has been shown to promote HCC growth and metastasis [39]. A carboxyl-terminal truncated HBx (Ct-HBx) is often observed in HCC tumor tissues. The DUB expression profiles of cells overexpressing Ct-HBx showed decreased levels of USP16, suggesting that the expression of USP16 may be negatively regulated in a manner dependent on Ct-HBx [40] (Figure 4B). Down-regulation of USP16 in the liver tumor cell line promotes its colony formation and tumor growth, leading to stem-like features. USP16 overexpression abolishes the tumorigenic ability of the Ct-HBx. USP16 was found to be frequently downregulated in human HCCs and correlates with advanced tumor stages and disease progression [40]. While this study identifies USP16 downregulation as a critical regulator in Ct-HBx-driven HCC growth, the targets of USP16 and its mechanism in this disease remain unclear.

### 2.3. Cytoplasmic and Nuclear Subcellular Localization of USP16

As noted in previous studies, USP16 is predominately localized to the cytoplasm during interphase and is in a hypophosphorylated state [13,17]. However, SDS-PAGE analysis of purified USP16 isolated from *Spodoptera frugiperda Sf9* cells was instrumental in identifying a slow-migrating proteoform that could be relinquished upon phosphatase treatment. Phosphorylation of Ser552 (pSer552) by cyclin-dependent kinase 1 (CDK1) was established to be the major modification [17]. While pSer552 is not required for tetramer formation, DUB activity, the substrate specificity of USP16, or transcriptional regulation by USP16, it is essential for cell proliferation and cell cycle progression through the G2/M phase. The underlying mechanism is the disruption of the interaction between USP16 and the nuclear export protein CRM1 (chromosomal maintenance 1, also known as exportin 1), resulting in its nuclear localization. Thus, Ser552 functions as the phosphorylation site regulating the nuclear retention of USP16 when cells enter the M phase of the cell cycle [17].

Furthermore, an NES between a.a. 572–581 was shown to contribute to the cytoplasmic localization of USP16 [21]. Whereby, following mitosis, USP16 is rapidly exported from the nucleus to the cytoplasm. As well, a non-canonical NLS sequence, a.a. 437–459, overlapping the predicted monopartite NLS signal, a.a. 439–456, was identified in USP16 [21]. Considering the predominant cytoplasmic localization of USP16, this NLS is weak. Enforced nuclear localization of USP16 abolishes DNA double-strand break (DSB) repair, possibly due to the unrestrained DUB activity [21]. This may link to the reduced DNA damage repair ability in Down syndrome cells [17]. Besides deubiquitinating H2Aub, recent studies also identified many cytoplasmic and nuclear substrates of USP16 (Figure 4 and Table 1). Below, we summarize these studies, highlighting the diverse regulation of USP16 in physiological and pathological processes.

## 3. The Substrates and Functions of USP16

### 3.1. USP16 as a Histone H2A DUB

#### 3.1.1. H2A and Gene Expression

One of the well-characterized functions of USP16 is acting as a histone H2A DUB (Figure 4C). Through an unbiased biochemical purification, USP16 was identified as the dominant DUB for histone H2A in vitro [14]. The function of USP16 in H2A deubiquitination in vivo was demonstrated by RNAi-mediated knockdown [14], knockout in mice [19], and overexpression and knockdown in the Down syndrome mouse model and cell lines [18]. ChIP-seq studies revealed that nuclear USP16 is predominately localized at gene promoter regions, and the knockout of the *Usp16* gene results in an increase of H2Aub at those sites, leading to increased gene repression [19,52]. In quiescent lymphocytes, USP16 also plays an essential role in regulating gene transcriptional activation, a process that is facilitated by Aurora kinase B-mediated phosphorylation of histone H3 at Ser28 [52]. These studies identified USP16 as the H2A DUB that antagonizes the histone E3 Ub ligase activity of PRC1 to regulate gene expression. We summarize the cellular functions governed by USP16-mediated gene expression below.

#### 3.1.2. H2A and Cell Fate Determination

USP16 is indispensable for mouse embryonic development as *Usp16* deletion is lethal to mice after implantation (embryonic day E3.5) but before gastrulation (E7.5), when cell lineage commitment starts [19]. Consistent with these observations, *Usp16* is not required for embryonic stem cell (ESC) viability but is required for differentiation [19]. This phenotype is similar to many epigenetic regulators [53]. The failure of *Usp16^−^*^/*−*^ ESCs to differentiate is possibly due to the inability of these cells to activate the expression of lineage-specific genes repressed by H2Aub [19]. The deubiquitination of H2Aub by USP16 is also a prerequisite for zygotic genome activation during the maternal-to-zygotic transition in mouse oocytes [41]. Conditional knockout of *Usp16* in oocytes caused defective zygotic genome activation and loss of developmental competence after fertilization. This defect largely resulted from H2Aub in zygotic genomes, which failed to be removed in the absence of *Usp16* [41]. Conditional knockout of *Usp16* in mouse hematopoiesis system causes mortality within two weeks, likely due to ammonia (preliminary observation). *Usp16* is found to be critical for the differentiation of hematopoietic stem cells into downstream progenitor cells [20]. Interestingly, these studies revealed that the defects in hematopoietic stem cell differentiation caused by the deletion of *Usp16* are due to the effects of USP16 on cell cycle progression, which can be rescued by the knockdown of the p53 target gene *Cdkn1a (p21)* [20]. These studies establish USP16 as a histone H2A DUB that controls cell fate during development.

#### 3.1.3. H2A and Somatic Stem Cell Maintenance

The function of USP16 in histone H2A deubiquitination was also supported by overexpression studies [18]. In humans, *USP16* is on chromosome 21, which is triplicated in Down syndrome. The potential contribution of USP16 to this developmental disorder was demonstrated with a humanized mouse model. Triplication of *Usp16* was found to reduce the self-renewal of hematopoietic stem cells and the expansion of mammary epithelial cells (by inhibiting Wnt signaling), neural progenitors, and fibroblasts [18,42]. Triplication of *Usp16* may also deubiquitinate H2Aub at gene promotors and activate the *Cdkn2a* gene, a critical regulator for cell senescence [18,33,34] (Figure 4C). Thus, USP16 levels may be linked to stem cell defects in the Down syndrome mouse model, although the relevance of this discovery to human Down syndrome is unknown.

#### 3.1.4. H2A and Neural Precursor Cell Function and Memory

Through its effects on gene expression, USP16 may also affect cell senescence (Figure 4C). Using mouse models of Alzheimer’s disease (AD) and human fetal cells with mutant amyloid precursor protein, dysfunction of cell-intrinsic neural precursor cells (NPC) was found to occur before inflammation and amyloid plaque pathology [43]. Downregulation of USP16 or its target gene *Cdkn2a* could reverse the impaired NPC self-renewal and reduce the accompanying cognitive defects and astrogliosis. The downstream targets of USP16 were identified by RNA sequencing, and surprisingly, the BMP signaling pathway was the most affected [43]. Hence, USP16 may represent a novel target to relieve the NPC defect and restore memory loss.

#### 3.1.5. H2A and EGFR-*lncEPAT* Pathway

The regulation of gene expression by the USP16-H2Aub pathway also operates during the tumorigenic process (Figure 4C). *LncEPAT* is a previously uncharacterized oncogenic long non-coding RNA (lncRNA). The EGFR pathway induces the expression of *lncEPAT*, for which high levels correlate with high glioma grade and poor patient survival [44]. Mass spectrometry identified USP16 as a *lncEPAT* binder, and it appears that *lncEPAT* blocks the recruitment of USP16 to chromatin, which in turn blocks USP16-mediated H2A deubiquitination and target gene expression (Figure 4C). Knockdown of *lncEPAT* enhances USP16-induced cell cycle arrest, cell senescence, and repressed tumorigenesis [44]. This study identifies USP16 as a critical regulator for the EGFR-*lncEPAT* pathway in glioblastoma tumorigenesis.

#### 3.1.6. H2A and DNA Damage Response

During DNA damage, histone H2A is ubiquitinated at Lys13/15 by the Ub ligase RNF168 [15]. This event plays a critical role in the recruitment of downstream DNA repair machinery to the double-strand break sites [54]. Through mass spectrometry, the HERC2 protein, a HECT domain Ub ligase, was identified as an interactor of USP16 [55]. The interaction region was further mapped to the C-terminal HECT domain of HERC2 and the disordered USP16 linker between ZnF-UBP and CD [16]. Possibly through this interaction, USP16 is recruited to DNA damage foci. Overexpression of USP16 negatively affects DNA damage-induced ubiquitinated foci, which may delay or reduce the recruitment of downstream factors. In this case, USP16 can also deubiquitinate H2AK15ub and fine-tune the Ub signal [16,21] (Figure 4D). Besides, USP16 was also found to be recruited to regions flanking DNA damage sites. USP16 deubiquitinates H2AK119ub *per se* and represses transcription, allowing the repair to be complete prior to the resumption of transcription [56]. Therefore, USP16 may coordinate the successful repair of damaged DNA.

### 3.2. USP16 as a DUB for Other Nuclear Proteins

#### 3.2.1. PLK1 and Chromosome Alignment

In early mitosis, the serine/threonine kinase PLK1 (polo-like kinase, also named serine/threonine-protein kinase 13, STPK13) is localized to the kinetochore. It regulates attachment of microtubules, which is critical for proper chromosome alignment [57]. Once an attachment is achieved, PLK1 must be removed from kinetochores, and this process is regulated by ubiquitination. Mass spectrometry identified USP16 as a PLK1 interacting protein that may regulate its deubiquitination [45]. In agreement with this, CDK1-mediated phosphorylation of USP16 increases its interaction with PLK1, and USP16 knockdown increases the ubiquitination level of PLK1 and decreases its association with kinetochores, which could yield improper chromosome alignment [45] (Figure 4E). The regulated interaction between USP16 and PLK1 kinetochores may cooperate with its regulation on global H2A mitotic deubiquitination to ensure proper chromatin alignment and segregation during the mitotic phase [45,58].

#### 3.2.2. C-Myc and Prostate Cancer

C-Myc, a well-known oncogene, plays important roles in many cancer types [59]. In a study with prostate cancer, USP16 was found to be positively correlated with the c-Myc signature [46]. Knockdown of *USP16* in prostate cancer cell lines reduces cell proliferation and suppresses xenograft tumor growth. USP16 was found to deubiquitinate c-Myc and regulate its levels in a manner dependent on DUB activity. The effects of USP16 on prostate cell growth are mediated by c-Myc, as the effect is enhanced in c-Myc down-regulated cells but diminished when c-Myc is overexpressed. Consistently, the level of USP16 is elevated in human prostate cancer compared to normal prostate tissues [46]. Collectively, USP16 may serve as a diagnostic marker and treatment target for c-Myc-driven tumor growth (Figure 4F).

### 3.3. The Substrates and Functions of USP16 in the Cytoplasm

#### 3.3.1. Calcineurin A and Immune Response

Calcineurin, also known as protein phosphatase 3, is a calcium and calmodulin-dependent serine/threonine phosphatase [60]. Calcineurin dephosphorylates the nuclear factor of activated T cells (NFAT), a family of transcription factors important for immune response. This results in its translocation into the nucleus. Activated NFAT then activates interleukin 2 expression and stimulates the growth and differentiation of T cell response [60]. However, in this cascade immune response, it is not clear how calcineurin-mediated NFAT dephosphorylation is regulated. Calcineurin is composed of two subunits: calcineurin A, a 61-kDa calmodulin-binding catalytic subunit, and calcineurin B, a 19-kD Ca^2+^-binding regulatory subunit [60]. Calcineurin A was found to be constantly poly-ubiquitinated at Lys327 and subsequently deubiquitinated by USP16 (Figure 4G). Deubiquitination of calcineurin A facilitates interaction with and dephosphorylation of NFAT, resulting in the activation of its downstream target genes. T cell-specific inactivation of *Usp16* prevents calcineurin A deubiquitination and NFAT activation, and mice suffered from less severe experimental autoimmune encephalitis and inflammatory bowel disease [47]. This study identifies USP16 as the DUB responsible for maintaining the activated status of calcineurin A (Figure 4G).

#### 3.3.2. IGF2BP3 and Gallbladder Cancer

The lncRNA *MNX1-AS1* (motor neuron and pancreas homeobox 1 antisense RNA 1) is an antisense RNA for *MNX1* [61]. *MNX1-AS1* has been implicated in many cancer types, including gastric cancer, colorectal cancer, and cholangiocarcinoma [61]. To address the role of *MNX1-AS1* in gallbladder cancer, it was found to enhance the binding of IGF2BP3 (Insulin-like growing factor 2 mRNA-binding protein 3) to USP16 [48] (Figure 4H). USP16 then deubiquitinates poly-ubiquitinated IGF2BP3 and protects it from proteasome-dependent degradation. Thus, USP16 stabilizes IGF2BP3, which may be the partial mechanism of *MNX1-AS1*-driven gallbladder cancer [48] (Figure 4H).

#### 3.3.3. JAK1/p38 Signaling and Lung Tumorigenesis

K-RAS mutations are oncogenic drivers in ~30% of lung adenocarcinoma (LUAD) [49]. Current therapy targeting LUAD with K-RAS activating mutations remains unsatisfactory, and further investigation into mechanisms regulating K-RAS-driven tumorigenesis could be beneficial to LUAD treatment. USP16 was then found to promote K-RAS-mediated lung tumors, and knockout of USP16 in mice attenuates lung tumorigenesis induced by *K-ras^G12D^* mutation. USP16 inhibits reactive oxygen species (ROS)-induced p38 activation that would otherwise be tumor suppressive. As well, USP16 deubiquitinates and stabilizes JAK1, and augments JAK1 signaling, hence promoting lung tumorigenesis (Figure 4I) [49].

#### 3.3.4. Low-Density Lipoprotein Receptor and Coronary Artery Disease

Low-density lipoprotein (LDL) particles are closely linked to atherosclerosis and coronary artery diseases [62]. LDL is taken up by membrane scavenger receptors, including LDL receptor (LDLR) [63]. Through screening the siRNA library against the USP family of DUBs, USP16 was found to affect LDL the most [50] (Figure 4J). Subsequent studies revealed that USP16 does not regulate the transcription of LDLR but regulates its protein stability. Co-immunoprecipitation revealed that USP16 interacts with LDLR and deubiquitinates the receptor, thus protecting it against degradation (Figure 4J). These data suggest that USP16 intervention could be a potential strategy for ameliorating atherosclerosis and coronary artery diseases [50].

#### 3.3.5. RPS27a and 40S Ribosomal Subunit Biogenesis

Ribosomes are central to protein translation and biogenesis. Both the small 40S and the large 60S subunits are tightly controlled [64]. The biogenesis of ribosomes starts from the formation of immature pre-ribosomal subunits in the nucleolus and then becomes translation-competent mature ribosomes in the cytoplasm. The biogenesis of the small 40S ribosomal subunits involves a final process of the 18S rRNA and the release of biogenesis factors, including RIOK1 [64]. RIOK1 is a protein kinase required for the cleavage of rRNA and the dissociation of two biogenesis factors from the ribosome, NOB1, and DIM2. USP16 was identified as a late cytoplasmic 40S pre-ribosome-associated factor by mass spectrometry [31] (Figure 4K). The interaction is through the ZnF-UBP domain of USP16 and a helix (a.a. 436–460) in the insertion region that splits the USP domain. Knockdown of USP16 causes the increased monoubiquitination of RPS27a at Lys113 and affects the late-stage assembly of the 40S subunit (Figure 4K). Thus, this study identifies USP16 as a critical factor for 40S subunit maturation, extending the function of USP16 to translation regulation.

#### 3.3.6. Tektin and Male Infertility

Tektin is a family of cytoskeletal proteins forming the outer doublet of microtubules [65]. Tektin heterodimer, along with axial periodicity matching tubulin, forms longitudinal polymers [65]. It is expressed in the centrioles and basal bodies of the male germ cell lineage and may be required for sperm activity and male fertility [65]. USP16 was found to deubiquitinate Tektin and prevent it from proteasome-mediate degradation (Figure 4L). Thus, USP16 may regulate male fertility by deubiquitination and stabilization of Tektin, ensuring the proper formation of microtubule filaments during spermiogenesis [51].

## 4. Conclusions

The balance between ubiquitination and deubiquitination is critical for homeostasis and vital to almost all cellular processes [4]. USP16, a member of the USP family, was first characterized as a histone DUB specific for histone H2A by removing the Ub moiety from Lys119 [14]. As a histone H2A DUB functioning in the nucleus, USP16 regulates gene expression [14,18,19], cell cycle progression [14,17,20], embryonic and somatic stem cell differentiation [18,19,20], zygotic genome activation [41], senescence [18,43], and the DNA damage response [16,21,56]. Nuclear USP16 also deubiquitinates PLK1 for mitotic chromosome alignment and c-Myc in prostate cancer [45,46]. In the cytoplasm, USP16 deubiquitinates calcineurin for T cell maintenance [47], LDLR for coronary artery function [50], RPS27a for 40S ribosomal subunit biogenesis and maturation [31], and Tektin for male fertility [51]. USP16 also deubiquitinates proteins in oncogenic pathways, for example, IGF2BP3 in gallbladder cancer [48], whereas Ct-HBx inhibits the expression level of *USP16* in hepatocellular carcinoma [40], revealing the intrinsic link to cancer development. Besides tumorigenesis, another important discovery is the link between USP16 and immune response. USP16 not only deubiquitinates calcineurin A to facilitate its interaction of NFAT and activation of inflammation genes but also deubiquitinates IKKβ, a key kinase involved in p105 phosphorylation and activation [36,47]. Thus, USP16 is linked to both T cells and B cells in the immune system. Interestingly, its transcription is under the control of NF-κB, revealing a positive feedback loop [35]. We have summarized the various substrates and regulatory networks of USP16 in Figure 4 and Table 1.

We envision that the functions of USP16 in the nucleus and cytoplasm are intricately spatiotemporally regulated. For example, during cell immune response, cytoplasmic USP16 deubiquitinates calcineurin A, allowing it to dephosphorylate and activate NFAT (Figure 4G). Activated NFAT then enters the nucleus to activate the expression of genes involved in immune response, such as interleukin 2. During gene activation in the nucleus, USP16 is also required for the removal of the repressive marker H2Aub at various promoter regions to ensure proper gene activation (Figure 4C). We predict harmonious coordination will be demonstrated in all USP16-regulated cellular processes, for example, USP16-regulated H2Aub mitotic global deubiquitination and PLK1 localization, as well as others.

Despite the multiple substrates and interacting proteins identified for USP16 to date, the mechanistic studies towards USP16 are still limited. The canonical function of USP16 is acting as a histone DUB to antagonize PRC1, the H2A E3 Ub ligase. Although USP16 is clearly demonstrated to facilitate transcriptional regulation in this manner, BRCA1-associated protein 1 (BAP1), a member of the UCH family of DUBs, can also deubiquitinate H2Aub and regulate transcription, implying a degree of redundancy [66,67]. The relationship between USP16 and other reported H2A DUBs remains to be determined. The intrinsic links between USP16 and human cancers, as well as immune response and coronary artery disease, suggest that USP16 can be a potential therapeutic target. However, inhibition or knockout of the USP16 gene may cause the upregulation of other H2A DUBs, such as USP12, BAP1, and MYSM1 [68]. Small molecule inhibitors or activators that can specifically regulate USP16 without causing the compensation of other DUBs is an exciting avenue not yet explored.

## Figures and Tables

**Figure 1 cells-12-00886-f001:**
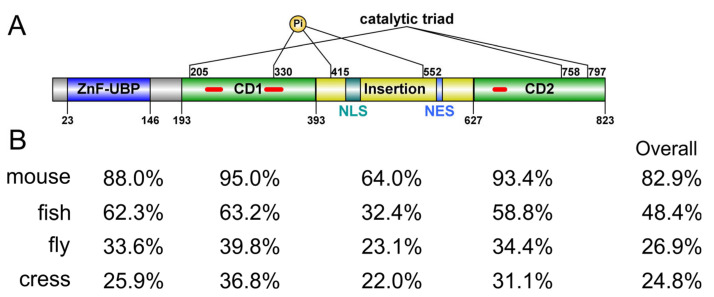
Domain architecture and conservation of USP16. (**A**) Sequence feature of human USP16. ZnF-UBP, UBP-type zinc finger domain; CD1/CD2, bifurcated catalytic domain; NLS, nuclear localization signal; NES, nuclear export signal; Pi, phosphorylation site. Short insertions in the catalytic domain are indicated with red lines. Domain boundaries are defined based on the predicted or experimental structures. (**B**) Sequence identity of the individual domains of USP16 and the full-length protein from different species (mouse: NP_077220; fly: NP_572220; zebrafish: NP_001139569; cress: NP_563719) compared to those of human USP16 (NP_006438). Pairwise sequence alignment was carried out using Smith-Waterman local alignment algorithm.

**Figure 2 cells-12-00886-f002:**
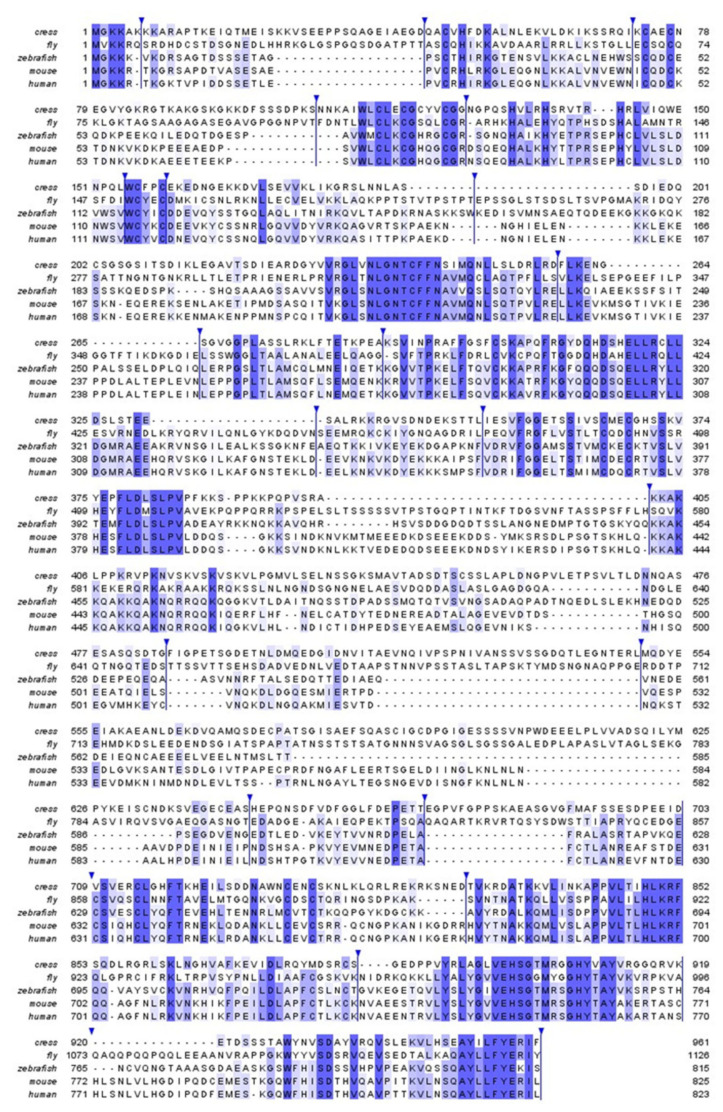
Sequence alignment of USP16 from five species. Sequences of human, mouse, zebrafish, fly, and thale cress USP16 homologs were aligned MAFFT algorithm. The sequences are colored according to percentage identity. Sequence fragments that exist only in one of the five homologs are removed from the alignment for clarity and indicated with a marker. The image was prepared using Jalview.

**Figure 3 cells-12-00886-f003:**
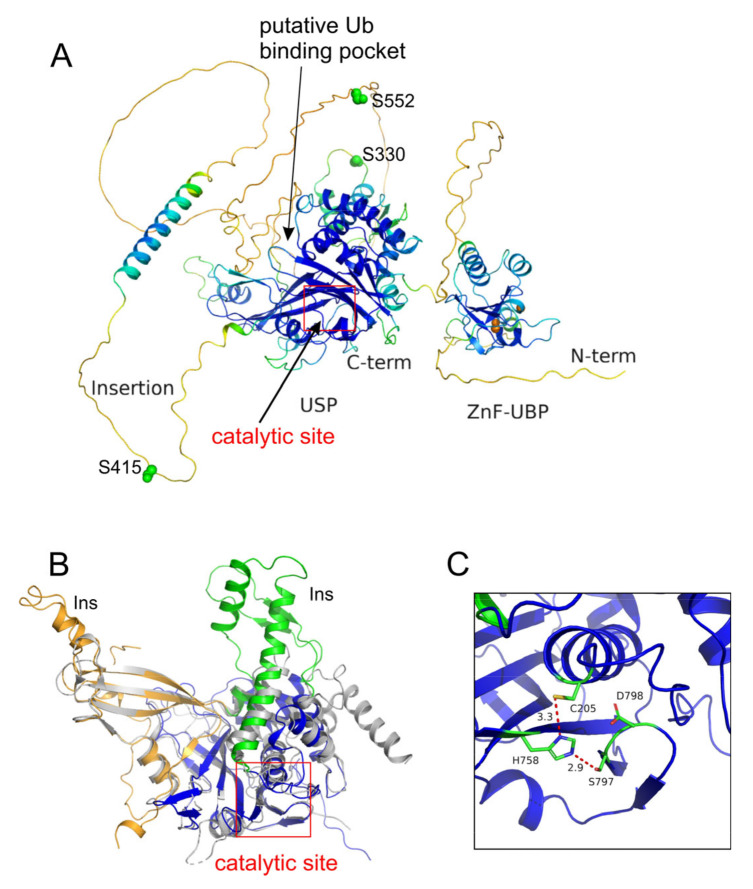
Structure of USP16. (**A**) AlphaFold-predicted structure of monomeric, full-length USP16 is colored according to the confidence of prediction (pLDDT value 0–100). Blue indicates high confidence. Zinc ions chelated to the ZnF-UBP domain are shown as orange spheres. Known phosphorylated residues, Ser330, 415, and 552, are presented as green spheres. (**B**) Superposition of the catalytic domains of USP16 (AlphaFold2 predicted) and USP7 (PDB: 1NB8) was carried out using PyMOL [29]. The fingers, thumb, and palm domain of USP16 are colored orange, green and blue, respectively. The USP7 catalytic domain is colored grey. While the core domains of USP7 and USP16 align well, it is evident that additional structural elements arise from the short insertions in USP16. (**C**) The catalytic triad of USP16. Residues C205, H758, and S797 form a hydrogen-bonded catalytic triad. D798, despite being close to the catalytic triad, is not arranged in the necessary conformation and, thus, is not a catalytic residue.

**Figure 4 cells-12-00886-f004:**
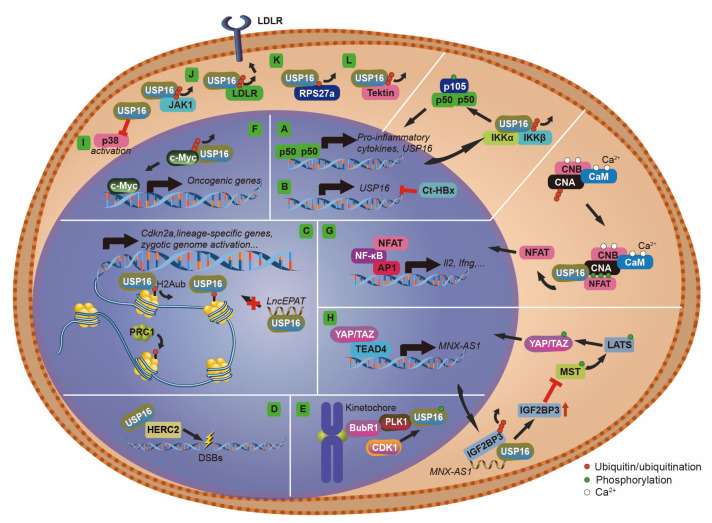
Substrates and functions of USP16 in the cytoplasm and nucleus. (**A**) Transcription of *USP16* is under the control of the NF-κB pathway, a pathway also regulated by USP16. (**B**) Ct-HBx inhibits the expression level of *USP16* in hepatocellular carcinoma. (**C**) USP16 functions as a major deubiquitinase of H2A, and *lncEPAT* binds to USP16 and blocks its recruitment to chromatin. (**D**) The C-terminal HECT domain of HERC2 interacts with the coiled-coil domain of USP16 and possibly recruits USP16 to DNA damage foci. (**E**) USP16 deubiquitinates PLK1 for mitotic chromosome alignment. BubR1 is a core component of the mitotic checkpoint complex. (**F**) USP16 deubiquitinates c-Myc in prostate cancer. (**G**) USP16 deubiquitinates calcineurin for T cell maintenance. CNA, Calcineurin A; CNB, calcineurin B; CaM, calmodulin; AP1, activating protein 1, a transcription factor. (**H**) USP16 deubiquitinates IGF2BP3 in gallbladder cancer. MST, mammalian STE20-like kinase; LATS, large tumor suppressor kinase; YAP/TAZ, Yes-associated protein 1 (YAP) and transcriptional coactivator with PDZ-binding motif (TAZ); TEAD4, TEA domain family member (TEAD) 4. (**I**) USP16 inhibits p38 activation and deubiquitinates JAK1 in K-RAS-driven lung tumorigenesis. (**J**) USP16 deubiquitinates the low-density lipoprotein receptor for coronary artery function. (**K**) USP16 deubiquitinates RPS27a for 40S ribosomal subunit biogenesis and maturation, and (**L**) USP16 deubiquitinates Tektin for male fertility.

**Table 1 cells-12-00886-t001:** Summary of the substrates and functions of USP16.

	Substrates	Functions	References
Nucleus	H2AK119ub	Deubiquitinates H2AK119ub and activates gene expression	[14,18,19,20,41,42,43,44]
	H2AK15ub	Deubiquitinates H2AK15ub in DNA damage response	[16]
	PLK1	Deubiquitinates PLK1 in chromosome alignment	[45]
	c-Myc	Deubiquitinates and stabilizes c-Myc and promotes prostate cancer cell growth	[46]
Cytoplasm	Calcineurin A	Deubiquitinates calcineurin A and regulates NFAT-targeted genes	[47]
	IGF2BP3	Deubiquitinates IGF2BP3 and promotes gallbladder cancer	[48]
	JAK1	Deubiquitinates JAK1 in lung tumorigenesis	[49]
	LDLR	Deubiquitinates LDLR in atherosclerosis and coronary artery diseases	[50]
	RPS27a	Deubiquitinates RPS27a for 40S subunit maturation	[31]
	Tektin	Deubiquitinates Tektin for microtubule filament formation during spermiogenesis	[51]
	IKKβ	Deubiquitinates IKKβ and activates NF-κB-targeted genes	[36]

## Data Availability

Not applicable.

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
