# Peer review of "The Pleiotropic Ubiquitin-Specific Peptidase 16 and Its Many Substrates"

_cells, 2023, doi:10.3390/cells12060886_

Round 1

Reviewer 1 Report

This is a well written and comprehensive review of the USP16 deubiquitinase. Authors discuss many aspects of USP16, including its structure, expression, substrates and functions. The USP16 structure prediction provides additional insights into its mechanisms. The paper will serve as a good review for readers interested in this important deubiquitinase.

Minor points:

1.      Line 82, missing “of” after “DUB”.

2.      Line 117, Figure 1A legends, the sentence is cut into 2 lines.

3.      Page 9, section 3A “USP16 functions as a histone H2A DUB”, the content in the 1st paragraph “Gene Expression” is redundant with section 2B “mechanisms regulating the mRNA and protein expression levels of USP16” (p6-p7) and can be consolidated with the paragraphs in 2B.

4.       Page 9-10, section 3A “USP16 functions as a histone H2A DUB” paragraphs could be better streamlined. The authors may consider consolidating discussion of diseases, e.g. “Down Syndrome” and “Glioblastoma”, in light of its functions as H2A DUB in cellular processes.

5.      With the many USP16 substrates and functions this review covers, it will be helpful to the readers to include a table summary listing USP16 substrates and the corresponding functions and references.

Reviewer 2 Report

The authors have been studying USP16 for many years and are leaders in this field. This review correctly cites most of the research papers on USP16. It provides a very well-organized summary of the molecular evolution, structure, regulation of expression and localization, physiological functions (regulation of gene expression, DNA repair, cell division, cell differentiation, translation, etc.), or pathological functions (involvement in cancer, inflammation, neurodegenerative diseases, etc.) of USP16. This reviewer has highly commended the paper and believes it will be accepted if the following issues are addressed.

Major comments:

1. On lines 102-103, the authors have assumed that UBP2 is an ortholog of human USP16 and UBP1 is an ortholog of USP45 based on slight differences in the similarity of the ZnF-UBP domains. However, the evidence seems weak. The authors should give other evidence.

2. On line 331, the correct description would be “which can be rescued by the knockdown of the p53 target gene Cdkn1a (p21) [17].”

3. On line 361, reference 40 seems incorrect. The other paper (PMID: 36197973) may be correct.

4. On line 435, is LDLR degraded by the proteasomal pathway? The cited paper (54) did not provide clear evidence. Some other reports show that LDLR can be endocytosed via ubiquitination and degraded by lysosomes.  

5. In Fig. 4A, the complex of p105 and p65 is incorrect, and this figure shows a wrong mixture of typical and atypical NF-kB pathways. 

Minor comments:

- On line 36, the authors should mention Met1-linked ubiquitination.

- On line 58, strictly speaking, the number of USP family proteins is ambiguous because USP17 has a lot of isoforms (PMID: 34454495).

- On line 62, please indicate the cited paper as the number (14?).

- On lines 80-85, each domain’s amino acid region differs from that in the UniProt database. Please cite papers defining each position.

- On line 96, please spell out PcG.

- On line 101 and Fig. 1B, the ZnF-UBP sequence identities between human USP16 and plant UBP2 are shown as 27.1% and 25.9%, respectively. Why are they different?

- On lines 103-105, the sentence’s meaning is ambiguous. Does it mean like this?: “These observations suggest the USP16 gene duplication event during the evolution of common ancestor of multicellular organisms, rendering USP16 fine-tuned for its functional roles”.

- On line 174, please indicate the PDB ID of the USP7 CD structure.

- On lines 235-238, the location of this sentence might be better in the “Gene Expression” section (lines 304-313).

- On line 280, please rewrite “Spodoptera frugiperda (Sf9)” to “Spodoptera frugiperda Sf9 cells”.

- On line 284, does the word “transcriptional regulation” mean transcriptional regulation of the USP16 downstream gene?

- The sentences “its deletion does increase the tendency of ESCs to differentiate (line 318)” and “knockout of 319 Usp16 causes the failure of these ESCs to differentiate (line 319” appear inconsistent. The authors should explain what “the tendency of ESCs to differentiate” means.

- On line 327, since there is no mention of ammonia in the original paper, it should be written as “likely due to ammonia (preliminary observation),” etc.

- On line 344, the subtitle “Senescence” seems unmatched because cell senescence is also discussed in other sections. The alternative is “Neural precursor cell function and memory” or like that.

- There were garbled characters in the distributed PDF file: lines 36, 226, 233, and 235 (maybe Greek characters).

- In Fig. 4, some molecules represented in the figure need to be explained in the main text or figure legend (BubR, AP1, MST, LATS, YAP/TAZ, and TEAD4). Or remove them from the figure if they are unnecessary.

- In Fig. 4B, Ct-HBx should be represented to regulate USP16 gene expression, not USP16 protein.

- In Fig. 4G, is AI autoinhibition region? CNA, CNB, and CaM also should be described as abbreviations in the main text or figure legend.

- In Fig. 4I, LDLR should be represented as a plasma membrane protein.

- The following articles on USP16 were not covered. The reviewer leaves it to the authors to decide whether to include them in this paper, but they would improve the coverage.

PMID 34294846: USP16 functions in the K-RAS-driven lung tumorigenesis through modulating p38 and JAK1 signaling.

PMID 30504774: USP16 modulates Wnt signaling in primary tissues through Cdkn2a regulation.

Reviewer 3 Report

Although originally the activity of the Ubiquitin-Specific Peptidase 16 (USP16) was restricted to the deubiquitination of the histone H2A, the experimental evidence collected in the last years indicates that the substrates range is broader has grown continuously over the years, including both cytoplasmic and nuclear proteins. Additionally, mechanistic insights regulating its activity has been elucidated, including its post-translational modifications. Zheng and coworkers in their manuscript titled "The Pleiotropic Ubiquitin-Specific Peptidase 16 And Its Many Substrates" review recent advances in USP16 field, emphasizing its potential role/s in neoplastic disorders and immune response. Actually, the review is well organized, comprehensive, and fluently written. I have just a few comments that should be addressed by the authors prior to the publication. 1) Since the authors discuss quite broadly the evolution of the peptidase I would consider seriously to replace the figure 2 (alignment) with a phylogenetic tree in which the authors display all the orthologs, including that of Spodoptera frugiperda. Regarding this specific issue, do the authors have any clue whether there are ortholog/s of USP16/USP45 in C. elegans, Nematostella vectiensis, Salpingoeca rosetta, Monosiga brevicollis and/or in lower eukaryote model organisms (e.g. P. polycephalum, Dictyostelia, Acanthamoeba, etc...)? And what about other plants other than A. thaliana (e.g. rice)? 2) A few typos are scattered throughout the main text and require to be carefully edited (e.g. lines 36, 117, 226, 233, 480 etc...). 3) Line 89: I would replace the term "homologs" with "orthologs". 4) To the reviewer it is not clear how the authors create the Figure 3B (superposition of the catalytic domain of USP16 and USP7). I would be extremely grateful if the authors could clarify it. 5) Eventually, from the manuscript it is not clear whether the USP16 subcellular localizations (cytoplasm versus nucleus) are mutually exclusive. And what about USP45? Does it display similar behavior? Could the author shortly discuss these issues as well.

Round 2

Reviewer 2 Report

This reviewer is satisfied with most of the authors’ responses but is still concerned about the molecular evolution of Usp16. Duplication of the ancestral gene of Usp16 and Usp45 might occur in the ancestor of vertebrates, resulting in Usp16 and Usp45, and independently in the ancestor of Arabidopsis, resulting in Ubp1 and Ubp2. In this case, it is nonsense to determine which Ubp1 or Ubp2 is an ortholog of human USP16. Another possibility is that gene duplication might occur in the common ancestor of multicellular organisms, resulting in Usp16 and Usp45, and that Usp45 was lost in the fly ancestor. If so, this would not be consistent with the authors’ claim that “Gene duplication events during evolution renders USP16 fine-tuned for its functional roles”.

The authors wrote, “We thus designate UBP2 as the Arabidopsis thaliana ortholog of human USP16, while UBP1 is likely the ortholog of USP45”, and “Such designation is purely based on bioinformatical analysis and requires futures experimental evidence for support.” However, based on the above reasons, this part should be rephrased as “We thus designate UBP1 and UBP2 as the Arabidopsis thaliana orthologs of human USP16”.

Reviewing the revised version, the reviewer found minor problems with the subheading, as indicated below. The authors should consider them before publishing the final version.

- Chapter 2A has a lot of topics. Subheading them will improve readability.

- Chapter 3 should be titled “The substrate and functions of USP16”.

- The subheads for topics within chapter 3A should be in style with 3B and 3C.

- An example of a subheading that improves the above issues is shown below. Underlines indicate the changes.

1. Introduction

2. Structure, expression regulation, and subcellular localization of USP16

2A. Protein structure of USP16

-Domain architecture and conservation

-Overall structure

-ZnF-UBP domain

-Catalytic domain

-Disordered regions

2B. Mechanisms regulating the mRNA and protein levels of USP16

-Gene Copy Number

-NF-κB (Nuclear factor kappa-light-chain-enhancer of activated B cells)

-Gene fusion

-HBV X (HBx) Protein

2C. Cytoplasmic and nuclear subcellular localization of USP16

3. The substrates and functions of USP16

3A. USP16 as a histone H2A DUB

-H2A and gene expression

-H2A and cell fate determination

-H2A and somatic stem cell maintenance

-H2A and neural precursor cell function and memory

-H2A and EGFR-LncEPAT pathway

-H2A and DNA Damage Response

3B. USP16 as a DUB for other nuclear proteins

-PLK1 and Chromosome Alignment

-C-Myc and Prostate Cancer

3C. The substrates and functions of USP16 in the cytoplasm

-Calcineurin A and Immune Response

-IGF2BP3 and Gallbladder Cancer

-JAK1/p38 signaling and lung tumorigenesis

-Low-Density Lipoprotein Receptor and Coronary Artery Disease

-RPS27a and 40S Ribosomal Subunit Biogenesis

-Tektin and male infertility

4. Conclusion

Author Response

We agree with the reviewer’s suggestions. Accordingly, we have rephrased the text as “We thus designate UBP1 and UBP2 as the Arabidopsis thaliana orthologs of human USP16” in Line 104-105.

We thank the reviewer for the constructive suggestions and have updated the subtitles accordingly.